# Influence of New Citrus Rootstocks on Lemon Quality

**Marlene G. Aguilar-Hernández [1], Lucía Sánchez-Rodríguez [2]** 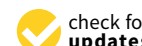**, Francisca Hernández [3]** ,
**María Ángeles Forner-Giner [4], Joaquín J. Pastor-Pérez [5] and Pilar Legua [3,*]**

1 Department of Horticulture, Universidad Nacional Agraria La Molina, Av. La Molina s/n, Lima 15026, Peru; maguilarhe@lamolina.edu.pe
2 Department of Agro-Food Technology, Research Group "Food Quality and Safety", Universidad Miguel Hernández de Elche, Escuela Politécnica Superior de Orihuela, Carretera de Beniel, Km. 3,2, 03312 Orihuela, Alicante, Spain; lucia.sanchez@goumh.umh.es
3 Department of Plant Sciences and Microbiology, Plant Production and Technology Research Group, Universidad Miguel Hernández de Elche, Ctra. de Beniel, Km. 3,2, 03312 Orihuela, Alicante, Spain; francisca.hernandez@umh.es
4 Citriculture and Vegetal Production Center, Valencian Institute for Agricultural Research, Apartado Oficial, 46113 Moncada, Valencia, Spain; forner_margin@gva.es
5 Departmento de Ingeniería Agroforestal, Universidad Miguel Hernández de Elche, Ctra. de Beniel, Km. 3,2, 03312 Orihuela, Alicante, Spain; jjpastor@umh.es
* Correspondence: p.legua@umh.es; Tel.: +34-966749669

**Abstract:** Citrus fruits are one of the main crops produced in the world with oranges, tangerines, lemons and grapefruits being among the most important. Among them, lemons are beneficial for human health because of their antioxidant activity, phenols and vitamin C. This study evaluates three rootstocks obtained in a Spanish breeding program (Valencian Institute for Agricultural Research (IVIA)): Forner-Alcaide 5 citrus rootstock ('FA 5'), Forner-Alcaide 13 ('FA 13') and Forner-Alcaide 517 ('FA 517') grafted onto cultivars 'Eureka´, 'Verna´, 'Fino 49' and 'Betera´. The results determined that rootstocks have influenced cultivars in terms of a decrease in the fruit size, weight, and thickness of the peel; but has increased the percentage of juice and soluble solids. With respect to phenol content and antioxidant activity, higher values were found with all cultivars studied on rootstocks 'FA 13', followed by 'FA 517'. Cultivars that were most influenced by rootstocks were 'Bétera' and 'Eureka' in physical properties, while in chemical properties they were 'Verna' followed by 'Fino 49'. This investigation demonstrated the interaction of rootstocks on different cultivars in morphological, biochemical and nutraceutical characteristics.

**Keywords:** Eureka; Fino; Forner-Alcaide; antioxidant activity

## 1. Introduction

Citrus fruits are one of the main crops produced in the world with oranges, tangerines, lemons and grapefruits among the most important. Production is located in the five continents North and South America, Europe, Africa, Asia and Oceania; but the largest citrus zone is located between 30° and 43° latitude of the Northern Hemisphere (United States, Mediterranean Basin and Japan) and in the Southern Hemisphere in Brazil, Argentina, South Africa and Australia. Lemon and lime worldwide production in 2017 was 17,218,173 t [1] and in Spain, it was 923.2 t [2], being the main citrus area Valencia, with oranges, tangerines and a lower proportion of lemons [3].

Different cultivars and rootstocks have mainly contributed to the development of the crop for its advantages and the economic success of citrus plantations [4], for example: rootstocks affect water relationships by influencing fruit growth, soluble solids and acidity [5], yield, weight fruit, juice percentage [6], and are used for threatening disease like Citrus Tristeza Virus (CTV) [7].

Prior to the arrival of CTV to Spain, sour orange (*Citrus aurantium* L.) was the standard rootstock in all plantations. Nowadays, in lemon orchards, *C. macrophylla* Wester is the only usable rootstock. It is tolerant to salinity, iron chlorosis and *Phytophthora* spp., but is susceptible to CTV and cold weather, and the fruit that it produced has low quality. Therefore, other rootstocks are being studied [4].

A citrus breeding program related to rootstocks started in 1974 at the Valencian Institute for Agricultural Research (IVIA). One of its purposes was to obtain rootstocks that grew better in Spanish conditions than rootstocks currently used [8]. A specific aim of the program was to develop rootstocks tolerant to salinity, iron chlorosis, water stress or flooding conditions [6]. Three rootstocks obtained from this breeding program are Forner-Alcaide 5 citrus rootstock ('FA 5'), Forner-Alcaide 13 ('FA 13') and Forner-Alcaide 517 ('FA 517'). The performance of these three rootstocks, grafted onto lemons, has been the subject of interest as a potential alternative to *C. macrophylla* which is very resistant to different abiotic plant stresses.

Citrus fruit have an excellent source of protective bioactive compounds for health [9]. The bioactive compounds of citrus fruits are involved in preventing carcinogenesis of the colon by being present in the juicy pulp or in the peel [10]. Cano and Bermejo [11] found that similar contents of bioactive compounds were different among species such as tangerine and orange, as well as in rootstocks.

The present study aimed to develop knowledge about three new citrus rootstocks Forner-Alcaide ('FA 5', 'FA 13' and 'FA 517') on the effect on the yield and fruit quality on four commercial cultivars of lemon: 'Bétera', 'Verna', 'Fino 49' and 'Eureka'. The quality was studied by evaluating their (i) morphology and physico-chemical parameters, (ii) chemical composition, and (iii) antioxidant properties in order to value their use for human consumption and/or obtaining biochemical compounds for other uses.

## 2. Materials and Methods

### 2.1. Plant Material, Experimental Conditions

'FA 5', 'FA 13' and 'FA 517' were the three citrus rootstocks used in the experiment. Seeds were collected from identified trees, of germplasm collection at IVIA, later grafted on four commercial lemon cultivars ('Bétera', 'Verna', 'Fino 49' and 'Eureka'). Cultivar was achieved through chip grafting and after one year it was transplanted to the final plot. Trees were evaluated when they were 15 years old. It is important to mention that, due to illness and water stress, 'Eureka' cultivar on 'FA 5' and 'Fino 49' on 'FA 13' did not prosper, with the consequent loss.

Rootstocks presented the following characteristics:

- 'FA 5' is a 'Cleopatra' mandarin and *Poncirus trifoliate* (L.) Raf. hybrid, characteristics are the size of the medium tree, branch presence of profuse thorns, short branches, trifoliate leaves, flat, small fruits, weight 35–36 g, diameter 40–45 mm, height 35 mm, thick rind. It presents CTV, *Phytophthora* spp. (disease) and *Tylenchulus semipenetrans* (nematode) resistance, limestone and flooding tolerant [12,13].
- 'FA 13', comes from same parents as 'FA5', but with some different characteristics: it induces smaller size to the grafted plants, resistant to salinity and susceptible to *T. semipenetreans.*
- 'FA 517' is a 'King' mandarin and *P. trifoliate* (L.) Raf. hybrid, with dwarfing characteristic, tolerant of limestone, salinity, CTV-resistant and nematodes [14].

'FA13' and 'FA517' rootstocks were registered in the European Union and they were commercialized (BOE 04/12/2007), while 'FA5' was registered before (BOE 10/29/2004) and all were obtained in 1978 [15].

Characteristics of these cultivars could be found as they were previously reported by IVIA [16]. Briefly, 'Bétera' is a fine lemon mutation, obtained in Valencia. The cultivar is not very productive and it is used mostly in gardening because it has few thorns. The fruit weight varies between 130 to 150 g and the percentage of juice is 30–40%. 'Eureka': obtained in California, the fruit is characterized by a weight of 120–140 g, percentage of juice 40–45%. 'Fino': very productive, sensitive to cold weather,

has two clones 'Fino 95' and 'Fino 49'. Fruit weight 110–130 g, percentage of juice 30–35%, and it has 2–6 seeds. 'Verna': of unknown origin, fruit weight 130–170 g, percentage of juice 30–35%.

Nursery work was undertaken according to the described by Legua et al. [6], plants grew at controlled conditions, anti-aphid mesh, temperature presented a mean of 22 °C and relative humidity was about f 80%. One year after being grafted, they were taken to the field.

The field was located at the IVIA Experimental Station (UTMX: 723368.000, UTMY: 4385233.000), Valencia, Spain. They were planted with a randomized complete block design, with 5 replications, the experimental unit was a grafted tree, and the distance between trees was 6 × 4 m. The edge effect was considered at the moment they were harvested. Soil characteristics were: clay loam soil, pH 8.5, $CaCO_3$ = 44.4%, active calcium carbonate 17.1% and electric conductivity (C.E.) 5.79 mS cm$^{-1}$. Cultural practices conducted manual pruning to third year, drip irrigation system, water characteristics (pH = 7.8, C.E = 2.0 − 3.5 mS cm$^{-1}$ and presence of boron from 400 to 500 mg kg$^{-1}$), fertilizer formula NPK = 4–1–1.5, and chemical weed control.

## 2.2. Morphology and Physico-Chemical Parameters

In the laboratory, 90 lemons were selected per cultivar/rootstock combination for analytical determinations. Lemons were not stored and were directly hand-squeezed. Three subsamples for each cultivar/rootstock combination (each one composed by 30 fruits) were randomly made, and then the fruits from each subsample were cut in half and carefully hand-squeezed in a commercial kitchen juicer. The freshly squeezed juices were centrifuged at 15,000× $g$ for 20 min (Sigma 3–18 K, Osterode and Harz, Germany) and were kept in freezer at temperature of −18 °C until analysis.

Fruit weight (FW) was determined with a Mettler balance model AG204 scale (Mettler Toledo, Barcelona, Spain). The size of each fruit: equatorial diameter (ED), polar diameter (PD), and peel thickness (PT) were measured, in each fruit, with a digital caliper model 500-197-20, 150 mm (Mitutoyo Corp., Aurora, IL, USA) on 30 fruits per replicate. pH, total soluble solids (TSS) and titratable acidity (TA) were analyzed as described by Forner-Giner et al. [8]. The maturity index (MI) was calculated as the ratio between the TSS and the TA.

Lemons peel color was assessed at four equidistant points of the equatorial region of individual fruits (the same 30 used for the measurement of ED, PD and PT) using a Minolta CR 2000 colorimeter (Minolta, Osaka, Japan), expressing the results using the CIE L* a* b* system (CIE, 1931). The mean values for lightness (L*), green-red (a*) and blue-yellow (b*) coordinates for each fruit were calculated. The objective color was calculated as chromaticity or chroma (C* = $(a^{*2} + b^{*2})^{1/2}$) and hue angle ($H^0$ = arctan (b*/a*)), and with them, the corresponding citrus color index (CCI) using the equation proposed by Jimenez-Cuesta et al. [17], CCI = 1000 a*/L* b*.

## 2.3. Sugar and Organic Acid Content

The extraction and quantification of sugar and organic acids in lemon juices were determined using three juice samples for each cultivar/rootstock combination as described by Legua et al. [18]. The juices were manually prepared by hand-squeezing in a commercial kitchen juicer and centrifuging at 15,000× $g$ for 20 min (Sigma 3–18 K; Sigma, Osterode am Harz, Germany). Then, 1 mL of supernatant was filtered through a 0.45 μm Millipore filter, and 10 μL was injected into a Series 1100 Hewlett-Packard high-performance liquid chromatograph (HPLC). A column SupelcogelTM C-610H (30 cm × 7.8 mm) and a pre-column (Supelguard 5 cm × 4.6 mm; Supelco, Bellefonte, PA, USA) were used for the analyses of sugars and organic acids. Organic acid absorbance was measured at 210 nm using a diode-array detector (DAD, HPLC, Waldbronn, Germany), while sugars were detected using a refractive index detector (RID, HPLC, Waldbronn, Germany). Standards of sugars and organic acids were obtained from Sigma (St. Louis, MO, USA) and calibration curves showed good linearity ($r^2 \geq 0.999$). Results for both individual organic acids and sugars were expressed g 100 mL$^{-1}$.

## 2.4. Antioxidant Activity (AA) and Total Polyphenol Content (TPC)

Extraction of antioxidants and total polyphenols was undertaken according to Wojdylo et al. [19]. This extract was used to measure three methods of antioxidant activity (AA) and total phenolic content (TPC). The free radical scavenging capacities were determined by three methods, ABTS$^+$ [20], DPPH$^\bullet$ radical [21], and FRAP [22]. Calibration curves (3.5–5.0 mmol Trolox L$^{-1}$) with good linearity ($r^2 \geq 0.999$) were used for the quantification of the AA by these methods. All analyses were done using an UV-visible spectrophotometer (Helios Gamma model, UVG 1002E, UK). Analyses were run in triplicate and results were expressed as mmol Trolox L$^{-1}$ of lemon juice.

TPC was method determined by Singleton et al. [23] and quantified using Folin–Ciocalteu reagent, as described by Gao et al. [24]. Gallic acid was used to prepare calibration curves. This analysis was run in triplicate and results were expressed as mg gallic acid equivalents (GAE) per 100 mL of lemon juice.

## 2.5. Statistical Design and Analysis

Normality test was conducted (results not shown) and, as a normal data distribution was obtained, one-way analysis of variance (ANOVA) and Tukey's multiple range test were performed to compare experimental data and determine significant differences among treatments ($p < 0.05$). Principal component analysis (PCA) using Pearson's correlation was also done. XLSTAT (2016.02.27444 version, Addinsoft) was used to perform all statistical analyses.

## 3. Results

### 3.1. Morphology and Physico-Chemical Parameters

The effect of rootstock and cultivars on morphology parameters can be found in Table 1. As can be observed, only the number of segments (NSG) did not show statistics differences among lemons under study. Fruit from 'Eureka' trees on 'FA 13' was significantly the heaviest (150.9 g) while lemons from 'Fino 49' trees on 'FA 517' presented the smallest weight with 95.5 g. With respect to 'Bétera' and 'Verna', contrary behavior was observed: on the first 'FA 517' presented higher weight while in the second, this rootstock produced lemons with smaller weight. Regarding ED, 'Bétera' on 'FA 5' was the highest (66.6 mm) and the lowest value was obtained with rootstock 'FA 517' in cultivars 'Verna' and 'Fino 49' with 56.3 mm and 54.6 mm, respectively.

**Table 1.** Effect of different rootstocks on lemons ('Bétera', 'Verna', 'Eureka' and 'Fino 49'), average fruit weight (FW, g), fruit equatorial diameter (ED, mm), fruit polar diameter (PD, mm), peel thickness (PT, mm), number of seeds (NSD), number of segments (NSG), volume of juice (VJ, mL) and percentage of juice (PJ, %).

| Cultivar | Rootstock | FW | ED | PD | ED/PD | PT | NSD | NSG | PJ |
|---|---|---|---|---|---|---|---|---|---|
| | 'FA 13' | $^\ddagger$ 116.7 bc | 58.0 bc | 77.2 bc | 0.7 cd | 4.92 ab | 8.7 bc | 9.6 | 39.3 cd |
| 'Bétera' | 'FA 5' | 119.2 bc | 66.6 a | 62.3 f | 1.0 a | 3.70 c | 8.0 bc | 9.3 | 48.9 a |
| | 'FA 517' | 126.0 ab | 58.5 bc | 70.6 cdef | 0.8 bc | 4.31 bc | 7.5 bc | 9.4 | 44.9 ab |
| | 'FA 13' | 138.7 ab | 64.2 ab | 75.3 bcd | 0.8 bc | 4.67 ab | 9.4 ab | 9.8 | 45.9 ab |
| 'Verna' | 'FA 5' | 137.9 ab | 59.3 bc | 90.6 a | 0.6 d | 4.82 ab | 5.8 c | 9.0 | 43.0 bc |
| | 'FA 517' | 114.5 bc | 56.3 c | 82.5 ab | 0.6 d | 5.26 a | 7.0 bc | 9.4 | 38.4 d |
| 'Eureka' | 'FA 13' | 150.9 a | 63.2 ab | 73.1 bcde | 0.8 bc | 4.42 abc | 7.3 bc | 9.1 | 45.0 ab |
| | 'FA 517' | 117.7 bc | 60.6 abc | 67.3 def | 0.9 b | 4.34 bc | 7.7 bc | 9.5 | 44.6 ab |
| 'Fino 49' | 'FA 5' | 126.7 ab | 60.6 abc | 71.6 cdef | 0.8 bc | 4.41 abc | 8.4 bc | 9.1 | 45.3 ab |
| | 'FA 517' | 95.5 c | 54.6 c | 63.7 ef | 0.8 bc | 3.5 c | 12.3 a | 9.7 | 45.5 ab |
| $^\dagger$ ANOVA | | *** | *** | *** | *** | *** | *** | NS | *** |

$^\dagger$ NS = not significant at $p < 0.05$; and ***, significant at $p < 0.05$, 0.01, and 0.001, respectively. $^\ddagger$ Values followed by the same letter, within the same column, were not significantly different ($p < 0.05$), according to Tukey's least significant difference test.

As EP measures the fruit length, it could be observed that the largest fruit sizes (90.65 mm) was on 'FA 5' from 'Verna' whereas the smallest fruit was found on 'Bétera' on 'FA 5' (62.32 mm). Fruit round shape (ED/PD) was 1.0 for 'Bétera' on 'FA 5' and was more elongated in 'Verna' on 'FA 5' (0.65). Peel thickness was thickest in 'Verna' on 'FA 517' with 5.26 mm and the thinnest was 'Fino 49' on 'FA517' (3.70 mm) and 'Bétera' on 'FA 5'. Fruit that had a greater number of seeds (NSD) were 'Fino 49' on 'FA 517' (12.3) and fewer seeds were found on 'Verna' on 'FA 5' (5.8). Percentage of juice (PJ) achieved the highest yield (48.9%) on 'Bétera' 'FA 5', and 'Verna' on 'FA 517' had the lowest yield (38.4%).

Color values of fresh lemon juices and peel are shown in Table 2. Regarding juice color, the highest value of L* was obtained for 'Verna' in 'FA 517' (40.41), which means the highest luminosity, while the lowest value was obtained for 'Bétera' on 'FA 5' (40.10), respectively. The a* value of lemon juice (red color positive values and green colors negative values) presented the greenest color for 'Verna' on 'FA 517' (−1.24) while the reddest color was found for 'Bétera' on 'FA 5' (−1.14) and 'Verna' in 'FA 13' (−1.15). Changes in b* values (blue color represented by negative values and yellow color by positive values) were found; 'Eureka', 'Fino 49' and 'Bétera' on 'FA 517' (3.29, 3.29 and 3.29, respectively) were similar statistically and the most yellow, while the lowest was 'Verna' on 'FA 5' (3.07). C* values were ranged from 3.5 to 3.3 for 'Bétera' on 'FA 517' and 'Verna' on 'FA 5', respectively. Hue* and CCI values in lemon juice samples, were statistically significant.

**Table 2.** Effect of different rootstocks on lemons ('Bétera', 'Verna', 'Eureka' and 'Fino 49') juice and peel, lightness (L*), red/greenness (a*), yellow/blueness (b*), chroma (C*), and hue angle (H°) values, and citrus color index (CCI = 1000 a*/L* b*).

| Cultivar | Rootstock | L* | A* | B* | C* | H° | CCI |
|----------|-----------|-----|-----|-----|-----|-----|-----|
| | | | | Juice Color | | | |
| | 'FA 13' | ‡ 40.36 ab | −1.20 bc | 3.13 bc | 3.36 bcd | 111.32 a | −9.73 b |
| 'Bétera' | 'FA 5' | 40.10 e | −1.14 a | 3.16 bc | 3.37 bcd | 110.13 b | −9.19 a |
| | 'FA 517' | 40.28 bcd | −1.20 bc | 3.29 a | 3.51 a | 110.37 b | −9.27 a |
| | 'FA 13' | 40.20 cde | −1.15 a | 3.14 bc | 3.35 cd | 110.43 b | −9.32 a |
| 'Verna' | 'FA 5' | 40.18 cde | −1.17 ab | 3.07 c | 3.30 d | 111.30 a | −9.79 b |
| | 'FA 517' | 40.41 a | −1.24 c | 3.21 ab | 3.46 ab | 111.33 a | −9.72 b |
| 'Eureka' | 'FA 13' | 40.15 de | −1.16 ab | 3.17 bc | 3.39 bcd | 110.37 b | −9.30 a |
| | 'FA 517' | 40.31 abc | −1.19 b | 3.29 a | 3.51 a | 110.30 b | −9.23 a |
| 'Fino 49' | 'FA 5' | 40.28 abcd | −1.16 ab | 3.20 ab | 3.41 abc | 110.31 b | −9.24 a |
| | 'FA 517' | 40.24 bcd | −1.18 ab | 3.29 a | 3.51 a | 110.08 b | −9.14 a |
| † ANOVA | | *** | *** | *** | *** | *** | *** |
| | | | | Peel Color | | | |
| | 'FA 13' | ‡ 68.81 e | 5.48 b | 54.81 cd | 55.17 de | 84.42 c | 1.43 c |
| 'Bétera' | 'FA 5' | 72.18 cd | 6.07 b | 55.60 bc | 56.01 cd | 83.84 c | 1.52 c |
| | 'FA 517' | 71.85 cd | 5.35 b | 52.47 e | 52.81 f | 84.32 c | 1.40 c |
| | 'FA 13' | 72.76 bc | 5.94 b | 57.30 b | 57.67 bc | 84.12 c | 1.43 c |
| 'Verna' | 'FA 5' | 72.65 c | 2.15 c | 52.17 e | 52.30 f | 87.80 a | 0.53 e |
| | 'FA 517' | 70.24 de | 3.49 c | 53.19 de | 53.43 ef | 86.45 ab | 0.89 de |
| 'Eureka' | 'FA 13' | 74.68 ab | 5.14 b | 56.85 b | 57.21 bc | 84.99 bc | 1.20 cd |
| | 'FA 517' | 73.20 bc | 8.75 a | 57.28 b | 57.91 abc | 81.35 d | 2.09 b |
| 'Fino 49' | 'FA 5' | 76.52 a | 5.87 b | 59.50 a | 59.85 a | 84.41 c | 1.29 cd |
| | 'FA 517' | 69.00 e | 9.70 a | 57.31 b | 58.21 ab | 80.38 d | 2.50 a |
| † ANOVA | | *** | *** | *** | *** | *** | *** |

† NS = not significant at $p < 0.05$; ***, significant at $p < 0.05$, 0.01, and 0.001, respectively. ‡ Values (mean of 3 replications for color juice and 12 replications for color peel) followed by the same letter, within the same column, were not significantly different ($p < 0.05$), according to Tukey's least significant difference test.

With respect to lemon peel color, in all the variables, color results were statistically significant ($p < 0.05$). The values for L*, both the highest and lowest, were found on 'Fino 49' cultivar but in

different rootstocks, on 'FA 5' was found the most luminous (76.52) while in 'FA 517' the less luminous with 69.0. Also 'Bétera' on 'FA 13' with 68.81 showed less luminous value. Regarding a*, cultivars 'Eureka' and 'Fino 49' on 'FA517' were the reddest peel colors (8.75 and 9.70, respectively) while the greenest was found on 'Verna' 'FA 5' (2.15). The highest values of parameter b*, was found on 'Fino 49' 'FA 5' which means the yellowish peel color, while 'Bétera' 'FA 517' and 'Verna' 'FA 5' were found as the blueness lemon peels. CCI 'Fino 49' on 'FA517' was the highest (2.50) and the lowest 'Verna' on 'FA 5' with 0.53.

The quality parameters, including pH, juice yield, TSS, TA, and MI of citrus juices are shown in Table 3.

**Table 3.** Effect of different rootstocks on lemons ('Bétera', 'Verna', 'Eureka' and 'Fino 49') fruit total soluble solids (TSS, °Brix), titratable acidity (TA, g citric acid $L^{-1}$), pH and maturity index (MI).

| Cultivar | Rootstock | pH | TSS | TA | MI |
|---|---|---|---|---|---|
| 'Bétera' | 'FA 13' | [‡] 3.09 | 8.66 de | 85.89 bc | 1.00 b |
| | 'FA 5' | 2.97 | 9.53 c | 97.41 ab | 0.98 b |
| | 'FA 517' | 2.88 | 10.73 a | 89.64 abc | 1.19 a |
| 'Verna' | 'FA 13' | 2.92 | 9.40 cd | 86.18 bc | 1.09 ab |
| | 'FA 5' | 2.92 | 8.43 e | 81.50 c | 1.04 ab |
| | 'FA 517' | 3.00 | 9.30 cd | 86.34 bc | 1.08 ab |
| 'Eureka' | 'FA 13' | 3.01 | 9.06 cde | 86.43 bc | 1.05 ab |
| | 'FA 517' | 2.91 | 10.70 a | 104.28 a | 1.02 ab |
| 'Fino 49' | 'FA 5' | 2.90 | 9.60 bc | 86.17 c | 1.11 ab |
| | 'FA 517' | 2.98 | 10.43 ab | 87.45 bc | 1.19 a |
| [†] ANOVA | | NS | *** | ** | ** |

[†] NS = not significant at $p < 0.05$; **, ***, significant at $p < 0.05$, 0.01, and 0.001, respectively. [‡] Values (mean of 3 replications) followed by the same letter, within the same column, were not significantly different ($p < 0.05$), according to Tukey's least significant difference test.

pH values did not show significant statistical differences. The rootstock 'FA 517' with cultivars 'Bétera', 'Eureka' and 'Fino 49' had the highest TSS (10.7, 10.7 and 10.4 °Brix, respectively), whereas 'Verna' on 'FA 5' had the lowest value (8.4 °Brix). TA had its maximum on 'Eureka' on 'FA 517' (104,28 g citric acid $L^{-1}$) and minimum on 'Verna' on 'FA 5' (81.5 g citric acid $L^{-1}$). In all cultivar/rootstock combinations, the maturity index ranged between values of 0.98 and 1.19.

## 3.2. Sugars and Organic Acids Content

Organic acids strongly influence the organoleptic properties of fruit. Table 4 shows sugars and organic acid profile of lemons under study. Citric acid and malic acid did not show significantly differences; whereas ascorbic acid concentration, showed slight variation, being 0.06 g 100 mL$^{-1}$ the highest concentration on 'Verna' on 'FA 517' and 0.02 g 100 mL$^{-1}$ the lowest on 'Eureka' on 'FA 13'. Succinic acid contents were found to be the highest (0.82 g 100 mL$^{-1}$) for 'Bétera' on 'FA517' and the lowest (0,47 g 100 mL$^{-1}$) for Eureka' on 'FA 13'. Regarding sugars, glucose and fructose were identified for all lemon samples. Both sugars showed a similar behavior in samples, being 'Bétera' on 'FA 517' the lemons with highest concentration, and 'Verna' on 'FA 5' presented the lowest values.

**Table 4.** Effect of different rootstocks on lemons ('Bétera', 'Verna', 'Eureka' and 'Fino 49') fruit citric acid (CA, g 100 mL$^{-1}$), malic acid (MA, g 100 mL$^{-1}$), ascorbic acid (AA, g 100 mL$^{-1}$), succinic acid (SA, g 100 mL$^{-1}$), glucose (Glu, g 100 mL$^{-1}$), fructose (Fru, g 100 mL$^{-1}$).

| Cultivar | Rootstock | CA | MA | AA | SA | Glu | Fru |
|---|---|---|---|---|---|---|---|
| | 'FA 13' | ‡ 3.61 | 0.36 | 0.04 abc | 0.60 abcd | 3.16 cd | 4.24 d |
| 'Bétera' | 'FA 5' | 3.83 | 0.46 | 0.04 abc | 0.59 bcd | 3.67 bc | 4.96 bc |
| | 'FA 517' | 3.88 | 0.57 | 0.04 abc | 0.82 a | 4.34 a | 5.59 a |
| | 'FA 13' | 4.00 | 0.44 | 0.03 bc | 0.69 abc | 3.79 ab | 5.00 bc |
| 'Verna' | 'FA 5' | 3.59 | 0.40 | 0.04 abc | 0.53 cd | 3.00 d | 4.09 d |
| | 'FA 517' | 3.79 | 0.52 | 0.06 a | 0.71 abc | 3.76 b | 4.94 bc |
| 'Eureka' | 'FA 13' | 3.48 | 0.39 | 0.02 c | 0.47 d | 3.59 bc | 4.78 c |
| | 'FA 517' | 3.88 | 0.54 | 0.04 abc | 0.77 ab | 3.91 ab | 5.34 ab |
| 'Fino 49' | 'FA 5' | 3.62 | 2.14 | 0.05 ab | 0.72 abc | 4.04 ab | 5.04 bc |
| | 'FA 517' | 4.11 | 0.51 | 0.05 ab | 0.76 ab | 3.98 ab | 5.32 ab |
| † ANOVA | | NS | NS | ** | *** | *** | *** |

† NS = not significant at $p < 0.05$; **, ***, significant at $p < 0.05$, 0.01, and 0.001, respectively. ‡ Values (mean of 3 replications) followed by the same letter, within the same column, were not significantly different ($p < 0.05$), according to Tukey's least significant difference test.

## 3.3. Antioxidant Activity (AA) and Total Polyphenol Content (TPC)

Antioxidants determined using different methodologies (FRAP (ferric reducing/antioxidant power), ABTS$^+$ (2,2-azinobis- [3 ethylbenzothiazolin-6-sulphoni]) and DPPH$^•$ (1,1-difenil2-picrilhidracil)) and total polyphenols of lemon cultivars under study are shown in Table 5.

**Table 5.** Effect of different rootstocks on lemons ('Bétera', 'Verna', 'Eureka' and 'Fino' 49) fruit total polyphenols content (TPC, mg GAE 100 mL$^{-1}$), and total antioxidant activity measured according to ABTS$^+$ assay (AA-ABTS+, mmol Trolox L$^{-1}$ fw), DPPH (AA-DPPH$^•$, mmol Trolox L$^{-1}$ fw), and FRAP (AA-FRAP, mmol Trolox L$^{-1}$ fw) content.

| Cultivar | Rootstock | TPC | AA-FRAP | AA-ABTS$^+$ | AA-DPPH$^•$ |
|---|---|---|---|---|---|
| | 'FA 13' | ‡ 320.5 ab | 1.12 bc | 4.20 | 1.21 |
| 'Bétera' | 'FA 5' | 284.6 b | 1.35 abc | 4.26 | 1.21 |
| | 'FA 517' | 327.3 ab | 1.51 abc | 4.47 | 1.02 |
| | 'FA 13' | 396.1 a | 1.58 ab | 4.18 | 1.52 |
| 'Verna' | 'FA 5' | 270.5 b | 1.32 abc | 4.73 | 1.37 |
| | 'FA 517' | 321.2 ab | 1.04 c | 4.07 | 1.91 |
| 'Eureka' | 'FA 13' | 245.9 b | 1.25 abc | 3.89 | 2.02 |
| | 'FA 517' | 340.9 ab | 1.25 abc | 4.82 | 1.52 |
| 'Fino 49' | 'FA 5' | 351.3 ab | 1.67 a | 4.42 | 1.43 |
| | 'FA 517' | 264.9 b | 1.35 abc | 4.57 | 1.60 |
| † ANOVA | | ** | ** | NS | NS |

† NS = not significant at $p < 0.05$; **, ***, significant at $p < 0.05$, 0.01, and 0.001, respectively. ‡ Values (mean of 3 replications) followed by the same letter, within the same column, were not significantly different ($p < 0.05$), according to Tukey's least significant difference test.

The two methods ABTS$^+$ and DPPH$^•$ did not show statistically significant differences. The highest value of FRAP was for 'Fino 49' on 'FA 5' (1.67 mmol Trolox L$^{-1}$) and lowest for 'Verna' on 'FA 517' (1.04 mmol Trolox L$^{-1}$) but no statistical differences were found among the other lemons under study. Concerning TPC, 'FA 13' rootstock had a different behavior on the cultivars, being the highest value (396.15 mg GAE 100 mL$^{-1}$) with 'Verna' and the lowest (245.91 mg GAE 100 mL$^{-1}$) with 'Eureka' although it was also found that 'FA 5' on 'Bétera' and 'Verna' showed statistically the lowest values, as well as 'Fino 49' on 'FA 517'.

### 3.4. Principal Component Analysis (PCA)

PCA representing cultivars and rootstocks and all parameters previously detailed are shown in Figure 1. This graphic, representing Pearson correlations among parameters, is a very useful statistical tool to established relationships among attributes that define samples characteristics. As it could be observed, this graphic explained 56.01% of data (x axis: 20.23%; y axis: 35.78%) and 3 groups were obtained. The first group correlates 'Verna' 'FA 5', 'Bétera' 'FA 13' and 'Verna' 'FA 517' due to H° peel and juice parameters, PD and PT. In the second group it could be found 'Fino 49'on 'FA 5', 'Bétera' on 'FA 5', 'Eureka' on 'FA 13' and 'Verna' on 'FA 13'. These lemons were correlated with FRAP and DPPH$^\bullet$ antioxidant activity methods, fruit weight, percentage of juice, equatorial diameter, ED/EP ratio, L*, b* and C* peel color, a* juice color and malic acid. Finally, the last group was formed by 'Eureka', 'Fino 49' and 'Bétera' on 'FA 517' rootstock and they are correlated with TPC, ABTS$^+$, number of seeds, number of segments, a* and CCI of the peel, C* and b* of juice, TA, MI, TSS, ascorbic acid, succinic acid, glucose and fructose.

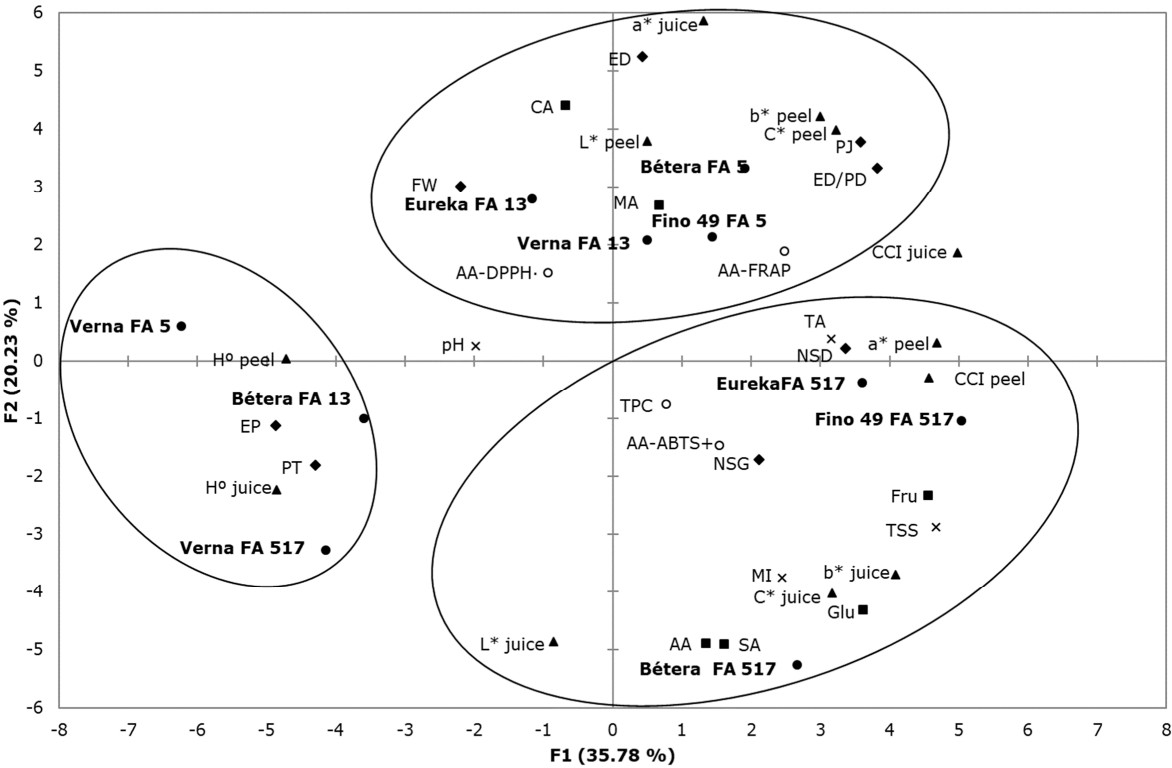

**Figure 1.** Principal component analysis (PCA) of variety and rootstocks (●), antioxidants and total polyphenols (°), morphological parameters (♦) (FW: fruit weight; ED: equatorial diameter; PD: polar diameter; PT peel thickness; NSD: number of seeds; NSG: number of segments; PJ: percentage of juice), color parameters (▲), chemical parameters (×) (TSS: total solid soluble; TA: titratable acidity; MI: maturity index) and organic acids and sugars (■) (CA: citric acid; MA: malic acid; AA: ascorbic acid; SA: succinic acid; Glu: glucose; Fru: fructose.)

## 4. Discussion

Citrus fruits are beneficial for human health because of their antioxidant activity, phenols and vitamin C; but they are also exposed to new diseases that appear in the citriculture as CTV [25]. The variables studied indicated that rootstocks have influenced cultivars in terms of decreases in the fruit size, weight, peel thickness; but have increased the percentage of juice and soluble solids; these data obtained differ from the aforementioned from IVIA [16] who reported higher values than those found.

Other research coincided with the current experiment for 'Eureka' on *C. macrophylla* that had smaller fruits, with higher content of titratable citric acid and sugars [26]; while Jiménez and Zamora [16] evaluated different rootstocks that had a higher values in weight (147.1 g) on *C. macrophylla*, diameter (60.8 mm), peel thickness (5.2 mm) in *C. volkameriana*, percentage of juice (33.6%) and soluble solids (8.7%) on Citrange 'Troyer'; acidity 6.43% in sour orange and maturity index Citrange 'Carrizo'. Different rootstocks influenced physical and chemical characteristics to fruits.

For 'Verna' cultivar, Perez et al. [26] indicated, during three years of evaluation, that weight and diameter results were not conclusive, but in other variables, some conclusions were obtained, for instance, further thickening of the bark, increase soluble solids, acidity and maturity index for rootstock *C. macrophylla*.

Deficit irrigation with saline water [27], showed a reduction in weight, peel thickness and TSS (total soluble solids) while juice percentage increased. These data agrees with our research where an inverse Pearson correlation was obtained ($r^2$ = –0.813, $p < 0.001$—data not shown) increasing the thickness of the crust, juice content decreases and soluble solids increase, detecting a negative correlation between these two variables ($r^2$ = –0.561, $p < 0.001$—data not shown). Liu et al. [28] indicated that differences in growth of fruit size is was related to several factors such as increase of cell expansion during the maturation period, high levels of auxin in fruits due to rootstocks or their progenitors, participation of multiple ARF genes, such as the ARF1 and ARF2 transcriptional repressors and ARF6 transcriptional activator, as well as the vigor of rootstock that develops an extensive root system absorbing water better; moreover, high levels of ABA (abscisic acid) can inhibit this mechanism, obtaining smaller size fruit.

Color is a subjective attribute that can be interpreted differently from one person to another, often defining purchase decision and relating it to the state of maturity or to the taste of fruit [29], and influencing commercial activity in the market [30].

The data obtained by measuring the skin shown that the rootstock have had an influence on the brightness (L*) of the fruits, mainly in the 'Fino 49' cultivar over 'FA 5´ and where the value of a* increases; but there is a high correlation (0.98) with the CCI. These tendencies are similar to those found by Legua et al. [6]. Thus, in other studies they pointed out that the color is due to the degradation of chlorophyll by the difference in temperatures, low temperatures with respect to average temperatures [31], to the color coordinates a*, b* and the angle index of hue with maturation [32]. The same team, Conesa et al. [33] worked with 'Eureka Frost', 'Lisbon Frost', 'Fino 49' and 'Verna 51' on *C. macrophylla* and their results indicated that the first three acquired the yellow coloration two months before 'Verna' which changed color at the end of January. Another influence in the values of the color is the time of harvest and the storage under average temperatures increasing these when fruits change of color [34].

Chemical parameters under study (pH, TSS, TA and MI) defined the quality of the fruit and determined the time of harvest, which have an influence on its post-harvest life. The results found in the present research indicated that the higher the value of soluble solids, the lower the acidity, and the maturity index increases in 'Bétera' and 'Verna' on 'FA 13' and 'FA 517' rootstocks, and these data coincide with other studies [35,36] where the increase in soluble solids was explained by the hydrolysis effect of the structural polysaccharides (pectins of the cell wall) in their basic components that would accumulate sugars [37]. It could also be due to the final stage of fruit development [38], accumulating mainly sucrose [39] and, when using organic fertilizer, decreasing the acidity but increasing the MI [40]. MI can also be increased by performing the scratching technique, as indicated by Soler et al. [41].

While other rootstocks differ in their influence as high acids, lower soluble solids and MI, similar results were reported in different studies [6,26,42–44]. The decrease in soluble solids with respect to other rootstocks could be because it is a non-climacteric fruit and has a respiratory activity where the oxidation of sugars would occur to obtain carbon dioxide, water and energy; other studies indicated that applications of ethephon in lemon decreased the content of soluble solids, acidity and MI [45]. The increase of acidity could be due to the inhibition of aconitase enzyme limiting the conversion of

citrate [46,47] indicated that the enzyme citrate synthase catalyzes the condensation reaction of acetate, which comes from acetyl-CoA and the oxaloacetate forming citrate, which decreases during maturation while increasing the activity of isocitrate dehydrogenase and succinate dehydrogenase. However, Soler et al. [41] points out that there are three genes that encode three aconitase enzymes, one in the mitochondria that manifests itself constantly in the process of development and maturation; the other two are located in the cytosol and at least one of them is manifested in the maturation; being the increase of the acidity directly related to a reduction in the level of transcription of the ATP citrate lyase, decreasing the degradation of citrate in the cytosol and leaving a greater amount of citrate stored in the vacuole.

Sugar and organic acid contents are major factors for fruit flavor and are important breeding traits. The major organic acid found in the present research was citric acid, which is also confirmed by other authors [45,48–50]. Citric and malic acid did not show significant differences. In contrast, with succinic acid and ascorbic acid were affected by rootstock, succinic acid has four carbon, is a dicarboxylic acid and is considered as a non-volatile acid [51] along with other components of different chemical nature participate in the taste properties of salty, astringent or bitter taste [52] and this would explain its low values.

Ascorbic acid (vitamin C) deficiency causes a disease known as scurvy that, according to Shaikh et al. [53], has other implications such as severe anemia with hemolysis (rash on neck, shoulders, chest, trunk and lower extremities). It is necessary to include in our diet citrus juice that has amount of ascorbic acid necessary for health; therefore, our results indicate that the rootstocks that had the highest value in ascorbic acid were 'FA 517' and 'FA 5' in all the cultivars and 'Eureka' had the lowest value of ascorbic acid on 'FA 13´ rootstock, this coincides with results reported by Ortiz et al. [54], where a similar behavior was found but with other varieties. Bermejo and Cano [55] found that 'Fino 49' lemon presented higher Vitamin C concentration on *C. macrophylla* rootstock.

Regarding sugars, results in the current research indicated that the rootstocks that had highest value in glucose and fructose were 'FA 517' and 'FA 5' in all the cultivars which differ from what was reported by Bermejo and Cano [55], who indicated that 'Fino 49' on *C. macrophylla* had values of glucose 4.36 g L$^{-1}$ and fructose 4.33 g L$^{-1}$ juice; while other research indicated that the content of fructose and glucose was higher in 'Fino 49' than 'Verna' [54] and this matches with results found in this study.

Citrus juice is a health benefit because it improves intestinal [56], cardiovascular [57], and cancer diseases [58]. These properties are related to its richness in phenolic compounds such as flavones and flavanones [59] and, within the latter group, erythrin and hesperidin are the main compounds in lemon juice [60]. In this research, the highest TPC was found on 'Verna' on 'FA 13' while 'Eureka' on 'FA 517' rootstock had a lower amount. Probably, other cultivars on other rootstocks do not have a similar behavior, as was observed for ascorbic acid, the reason why these data are not conclusive; this should be due to the fact that the content of phenols differs at the moment of harvest [61], in terms environmental conditions, and in the seasons [62]. Phenols also depends on the received brightness between plants [63], parts of plants (leaves, fruit), peel, pulp or juice, phenol compounds vary from one cultivar to another.

Concerning antioxidants, the results indicate for ABTS$^+$ and DPPH that they had little ability to inhibit free radicals and that this was similar between cultivars and rootstocks, being statistically not significant. In contrast to FRAP, where the highest ferric-reducing capacity was found on 'Verna' on 'FA 13' while the lowest was 'Eureka' on the same rootstock. The data shown differ from those found by other authors as [64] they point out that antioxidant activity in lemon varies significantly with different degrees of maturity depending on when they were harvested, being greater in green than in mature specimens. In summary, antioxidant activity will depend on many factors, but mainly on the amount of compounds present, because antioxidants have a similar structure, isomerism, stereochemistry and the lower or higher presence of flavone or flavonones.

## 5. Conclusions

In conclusion, results indicated that best rootstock was 'FA 517' that had a positive influence on cultivars studied as well as the physical variables of smaller fruits, thin rind, increase in soluble solids, higher maturity index and the content of organic acids and carbohydrates.

With respect to the content of phenols and antioxidant activity, higher values were found in all cultivars studied on rootstocks 'FA 13', followed by 'FA 517'. Cultivars that were most influenced by rootstocks were 'Bétera' and 'Eureka' in terms of physical properties while, in chemical properties, they were 'Verna' followed by 'Fino 49'. Summing up, each rootstock had an influence on the different cultivars studied in terms of the physical, chemical and antioxidant properties.

**Author Contributions:** M.G.A.-H. and L.S.-R. performed the experiments and wrote the manuscript; M.Á.F.-G. and J.J.P.-P. analyzed the data; F.H. and P.L. coordinated the study, planned and designed the experiments. All authors have read and agreed to the published version of the manuscript.

**Funding:** This research received no external funding.

**Acknowledgments:** Work of Aguilar was supported by a scholarship from Fundación Carolina.

**Conflicts of Interest:** The authors declare no conflict of interest.

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
