# Peer review of "Influence of New Citrus Rootstocks on Lemon Quality"

_agronomy, doi:10.3390/agronomy10070974_

Round 1
Reviewer 1 Report
In the paper entitled “Influence of new citrus rootstocks on lemon quality” Manuscript ID: agronomy-824598, the authors investigate the effect of different rootstock on the lemon quality in different cultivars. The main goal of this work was to study the effects in the lemon characteristics (morphological, biochemical and nutraceutical) of some cultivars in different rootstocks. The manuscript is novel and highly interested and the authors provide information related with the best cultivar that should be graft in the best rootstock for it. However, the English is not clear, and it should be revised by a native speaker in order to facilitate reading. Furthermore, the authors should review the rules of the journal and correct the bibliography. The work seems to have been carefully carried out but I have some doubts that need to be addressed:
In Keywords: It is better don´t put the same words that in the title
Throughout the text it is used varieties and patterns but is but it is more scientific to use cultivars and rootstock respectively.
In Introduction:
It would be necessary to talk about the characteristics chosen for the essay by expanding the penultimate paragraph because it makes no sense.
In material and methods:
Section 2.1. Perhaps it would be easier to see the characteristics of the rootstock and the cultivars if they were in a table.
It would be necessary to know the type of graft that has been performed, how old are the grafts and how long have they been in the field.
Section 2.5. Were the data tested for normality? and for homogeneity of variances with the Levene´s test?
In discussion:
The discussion perhaps it is too long and needs a review in the writing and use of English. For better understanding it should be written, shorter, and more concise. For example, the first paragraph is a single sentence and it would be better to use shorter sentences. This would apply to the all discussion.
Author Response
Please see the attchment

Reviewer 2 Report
Comments are given in manuscript PDF.
